# Design and Fabrication of the Vertical-Flow Bioreactor for Compaction Hepatocyte Culture in Drug Testing Application

**DOI:** 10.3390/bios11050160

**Published:** 2021-05-19

**Authors:** Liang Zhu, Zhenfeng Wang, Huanming Xia, Hanry Yu

**Affiliations:** 1Singapore Institute of Manufacturing Technology, Agency for Science, Technology and Research (A*STAR), 2 Fusionopolis Way, #08-04 Innovis, Singapore 138634, Singapore; zhu_liang@simtech.a-star.edu.sg (L.Z.); hmxia@njust.edu.cn (H.X.); 2Mechanobiology Institute, National University of Singapore, T-Lab, #05-01, 5A Engineering Drive 1, Singapore 117411, Singapore; zfwang@simtech.a-star.edu.sg; 3Institute of Biotechnology and Nanotechnology, Agency for Science, Technology and Research (A*STAR), The Nanos, #04-01, 31 Biopolis Way, Singapore 138669, Singapore; 4School of Mechanical Engineering, Nanjing University of Science and Technology, 200 Xiaolingwei St., Nanjing 210094, China; 5Department of Physiology, Yong Loo Lin School of Medicine, MD9-04-11, 2 Medical Drive, Singapore 117597, Singapore; 6Singapore-MIT Alliance for Research and Technology, 1 CREATE Way, #10-01 CREATE Tower, Singapore 138602, Singapore

**Keywords:** microfluidics, bioreactor, perfusion culture, hepatocytes, drug testing, design, fabrication

## Abstract

The perfusion culture of primary hepatocytes has been widely adopted to build bioreactors for various applications. As a drug testing platform, a unique vertical-flow bioreactor (VfB) array was found to create the compaction culture of hepatocytes which mimicked the mechanic microenvironment in vivo while maintaining the 3D cell morphology in a 2D culture setup and enhancing the hepatic functions for a sustained culture. Here, we report the methodology in designing and fabricating the VfB to reach ideal bioreactor requirements, optimizing the VfB as a prototype for drug testing, and to demonstrate the enhanced hepatic function so as to demonstrate the performance of the bioreactor. This device enables the modular, scalable, and manufacturable construction of a functional drug testing platform through the sustained maintenance of model cells.

## 1. Introduction

With the development of in vitro culture of hepatocytes since the 1980s, there have been numerous models and reactors created, catering to various applications such as the bio-artificial liver (BAL), drug toxicology testing, therapeutic development, and tissue engineering [1,2,3]. Due to high metabolism and detoxification properties of hepatocytes, it is preferable to use a perfusion culture system for adequate nutrition and oxygen supply as well as fast waste removal [4,5]. However, the shear stress generated by perfusion may damage the functionality or even the viability of hepatocytes [6,7]. An essential component of a perfusion bioreactor is therefore the shear stress filter, which minimizes the shear stress when the perfusate passes by the cells while still maintaining the level of mass exchange.

The 3D-μFCCS our laboratory has developed made the first step in achieving this balance by integrating pillar filters around the cell seeding channel so that the loading cells may be well trapped and so that the shear stress may be “filtered” out during the perfusion later on [6,8]. On top of that, there is an improved protocol by incorporating compaction force on top of perfusion culture to achieve better hepatocyte performances [9]. However, the problems associated with the pillar filter in a cross-flow perfusion—the mass transfer inefficiency [10], low throughput [8], uncontrollable manual compaction [9], limited accommodation of cells [9], etc., still remain unsolved. We have thus reported previously that the adoption of a nano-pore size membrane as the shear stress filter in a bioreactor allows for a buildup of pressure in the culture chamber, and leads to a “compaction” culture of the hepatocytes, which maintains a 3D morphology in a monolayer culture setup, enhancing the cell polarity and other hepatic functions [11]. This compaction culture is realized in a vertical-flow bioreactor (VfB) we made.

In order to showcase the methodology used to design, fabricate, and test this unique VfB as a drug-testing platform prototype, this paper elaborates on the principles of the design, the methods of the experimental setup and optimization, as well as the hepatic function test, while using the double-well VfB as a case study. Notably, the VfB utilizes modular culture chambers which are scalable, ranging from a single well reactor to a 24-well array, increasing its flexibility in adapting the throughput, and reducing the cost of the VfB-based drug testing product to fulfill different application requirements.

## 2. Materials and Methods

### 2.1. Bioreactor Design and Fabrication

The VfB was designed using Solidworks 2012 (Dassault Systèmes SolidWorks Corporation, Waltham, MA, USA) and AutoCAD (Autodesk, San Rafael, CA, USA). The computational fluidic dynamic (CFD) simulation was completed by ANSYS-CFX (ANSYS, Inc., Canonsburg, PA, USA). The main body of the VfB is fabricated by the thermoplastic polymethyl methacrylate (PMMA), which is biocompatible, transparent, and suitable for scaled-up production when the VfB transits to the industrialization stage [12,13]. Thermo-bonding was used for the assembly of the VfB, either by hydraulic pressure thermo-bonder (Atlas, Specac, Orpington, UK), or in a high-temperature oven (Modell 100–800, Memmert, Schwabach, Germany). Prior to bonding, the PMMA VfB layers were vacuum dried in a vacuum oven (VOS-301SD, Eyela, Singapore). All bonding work was done in a Class 1000 clean room in order to control the clean environment to ensure bonding strength.

The double well VfB has two independent chambers linked with perfusion tubing and valve fixed onto a PMMA jig. Each chamber of the VfB utilizes the same size of that in a commercially available 24-well culture plate (Nunc^®^, Thermo, Waltham, MA, USA). The vertical-flow perfusion design uses the nano-pore-size membrane in Polyester (PET, Transwell^®^, Corning, Corning, CA, USA) as both the shear stress filter and the cell seeding membrane.

### 2.2. Cell Seeding and Culture

Fresh primary hepatocytes were isolated from the male Wistar rats (250–300 g). William’s E medium was used as culture media, (Sigma, St. Louis, MO, USA) supplemented with 1 mg/mL bovine serum albumin (BSA) (PAA Laboratories, Pasching, Austria), 10 ng/mL epidermal growth factor, 0.5 mg/mL insulin, 5 nM dexamethasone, 50 ng/mL linoleic acid, 10 units/mL penicillin, and 10 mg/mL streptomycin. The hepatocyte suspension was gently mixed and pipetted onto the collagen-coated cell seeding membrane in a spot-by-spot way so as to make the cell suspension reside on the membrane evenly. The hepatocytes were then maintained either in perfusion, static, or collagen sandwich culture (as control) in the CO_2_ incubator with 5% CO_2_ and 95% humidity at 37 °C. For the sandwich culture, an overlay of 0.3 mg/mL collagen was added onto the cells the following day.

The animals used in the experiments were handled under the approval of the Institutional Animal Care and Use Committee (IACUC) of the National University of Singapore (NUS). No human samples or participants were used, and no other ethics approval is required.

### 2.3. Hepatic Function Test

Perfused culture media were collected daily for each chamber of the bioreactor and kept at −20 °C until the assay was carried out. The control was the supernatant of the static cultured hepatocytes on a 24-well culture plate. Urea production was quantified by BUN urea nitrogen test (Stanbio Laboratory, Boerne, TX, USA). The CYP level was quantified on the CYP specific metabolites, with the internal standards (Cyp1a2, Cyp3a2: 100 ng/mL Acetaminophen-D4; Cyp2b2: 50 ng/mL OHbupropion-D6), by LC/MS measurement (LC: 1100 series, Agilent, Singapore; MS: LCQ Deca XP Max, Finnigan, Singapore). All functional data were normalized to the seeded cell numbers, which were determined by the PicoGreen assay (Molecular Probes, Eugene, OR, USA).

## 3. Results and Discussion

### 3.1. Design and Principles

An ideal bioreactor for in vitro cell culture device, regardless of static or perfusion culture, should be constructed with nontoxic and biocompatible materials [14], preferably with optical transparency for cell status monitoring [15]. Each culture well or chamber needs to provide a large surface area for a sufficient number of cells to attach [14], high mass transfer efficiency [14], and preferably with flexible configuration to allow for co-culture, cell–cell signaling assays, etc. [15]. The culture device needs to be easy to assemble, allow for the easy insertion and retrieval of scaffolds or substrates, and with scalability for high throughput culture [15]. When it comes to a perfusion bioreactor, additional requirements arise such as providing a leak-proof culture chamber or well to allow for a medium flow perfusion with controllable flow types and rates to avoid the negative impact caused by the formation of air bubbles [15,16]. To maintain a good microenvironment of the cells under the perfusion culture, the perfusate needs to be evenly distributed across the large cell attachment surface, with sufficient culture medium replenished in-time and shear stress minimized [11]. Additionally, if the bioreactor is able to mimic the in vivo microenvironment during cell culture through maintaining the cells in 3D morphology or keeping the cells under a similar pressure as found in vivo, both the cell functions and the viability are expected to be enhanced and maintained for a sustained period of culture [11].

The VfB was designed to meet with the aforementioned features of an ideal perfusion bioreactor, with the unique “vertical-flow” perfusion to realize the compaction culture, and thus mimicking the in vivo microenvironment for cells [11] (Figure 1), while avoiding the concentration gradients of culture media caused by the cell adsorption in the front of the perfusion path in many commonly seen “horizontal-flow” microfluidic bioreactors [6,8,17] (Figure 1). If the perfusion path is horizontal across the culture chamber of 15 cm diameter, the media solvents will be absorbed by the cells along the path, resulting in a reduction of the media concentration while the flow length (and hence the cell-media contact time) increases [18] (Figure 1A). In contrast, the vertical-flow perfusion path only comes in contact with cells along the height of the cell monolayer (in μm level). The perfusate flow can then be designed to reach and leave the cells across the culture area simultaneously, which in theory should create an evenly distributed media supply in the original concentration (Figure 1B). There is also the mass transfer problem present in horizontal-flow (including cross-flow) reactors: the flow rate has to be carefully balanced to minimize shear stress and to optimize mass transfer, which in turn means that not all the perfusate will come into contact with the cells [19]. In the case of a vertical-flow reactor, the vertical flow pushes all the perfusate onto the cells, which maximizes the media supply to the cells with shear stress kept at a minimum. Moreover, the vertical-flow design’s culture area is easily scaled up because the perfusion is no longer made on 2D surfaces unlike many microfluidic chips [6,15,18,20]. All perfusion channels and tubing would be located above and below the culture chambers so that the footprint may be compact enough to accommodate multiple wells with sufficient well sizes, enabling the vertical-flow bioreactor design to be modular and scalable.

This VfB also adopts the open-cap design for the ease of cell seeding and harvest, therefore facilitating downstream assays such as staining, imaging, RNA extraction, etc. The cap of VfB was constructed with two layers of micro-featured PMMA with a membrane filter sandwiched via permanent bonding, and the bottom consists of another two slayers of PMMA structure to form the culture chambers (Figure 1B and Figure 2F). In this setup, the cell culture membranes can be reversibly fixed into individual culture chambers, via ring clips (Figure 3B). The VfB was then fixed into the PMMA jig, with a silicone ring to seal the interfaces between the rigid PMMA parts (Figure 3A,C), and with tubings and valves connected to the jig (Figure 3D).

Layer 1 of the cap and Layer 2 of the bottom is the “flow enlarger”, which interfaces with the Φ1 mm (ID) perfusion tubing and the Φ15 mm membrane-filter. The entry orifice is a Φ1 mm through hole, linking to an inverted V shape structure with 8 guiding grooves on the inner wall, so that the flow could be easily wicking along the wall and fills the whole shallow enlarger chamber (Figure 1B and Figure 2F).

Layer 2 of the cap and Layer 1 of the bottom contains the “flow distributor” structure, which is an array of through holes with the diameters at 0.400, 0.480, 0.576, 0.691, 0.829, 0.995 mm from center to periphery (Figure 2D). The numbers were optimized via CFD simulation in ANSYS (Figure 2E).

In our consideration in coming up with a good bioreactor design, we have summarized the ideal features of cell culture devices, especially bioreactors capable of perfusion culture and with biometic features mimicking the in vivo microenvironment of cells. We have then tried to create a VfB design to meet all these requirements (Table 1).

Along with the abovementioned considerations in Table 1, the key functions of a VfB are elaborated below.

The cap would carry out the following functions:
**Minimize the shear stress**: At a given flow rate, the expansion of the cross section area perpendicular to the flow direction may result in the decrease of velocity, thus lowering the shear stress at the exit of the expansion structure, which serves as the “flow enlarger” in the VfB. The “membrane filter”, sandwiched between the two layers of the cap, also serves as a shear stress filter, as the membrane was track-etched with micro-pores (Φ0.4 or 3 μm) so that the high resistance of these pores will force the perfusion flow to sieve through the membrane. This would lower the impact of perfusion flow onto the fragile cells below the filter, mimicking the in vivo condition of plasma sieving through blood vessel walls (e.g., pore size Φ0.4 μm on mice vessels) [21] (Figure 1B and Figure 3A).**Even distribution of flow across the whole cross section area**: As the diameter of the perfusion tubing is as small as Φ1 mm while the diameter of the membrane filter is expanded to Φ15 mm, a “flow distributor” is needed to dissipate the flow to the periphery of the culture chamber and force the flow velocity to distribute evenly across the area of the culture chamber below (Figure 2F). If the through hole array in the flow distributor were of the same size and arranged at the same intervals, the flow velocity of the media would be much faster at the center than at the sides (Figure 2B) when taking sampling points across the diameter (Figure 2A). It is therefore important to increase the flow resistance in the middle. The computational fluidic dynamic (CFD) simulation showed that the flow velocity was much more evenly distributed after this adjustment, other than the part for accommodating ring clips (Figure 2C versus Figure 2E).**Low buffer volume for in-time media replenishment**: As the cells in the perfusion path of VfB receives 100% perfusate due to the vertical-flow design, the flow rate is controlled at a much lower level than the horizontal or cross-flow bioreactor types due to the minimized loss of flow portion in parallel with the cell seeding plane. The volume before the cell substrate, considering as the “buffer volume” was then kept at a low level so that the media replenishment rate may still be maintained. Since the cell culture area needs to be large, the height of the cell culture chamber, which is the distance between the membrane filter and the cell seeding membrane, needs to be kept low. This is the reason why the culture chambers in VfB are shallow—to achieve a low buffer volume for in-time media replenishment.

The bottom of the VfB functions as the cell culture module, with the PET membranes reversibly fixed into the culture wells as the cell seeding substrates. The features of the VfB culture module include:
**Enough cell number per data point**: In prevalent microfluidic culture devices, the cell number per data point is commonly limited by the micro-chambers. For example, the 3D-μFCCS (3D microfluidic channel-based cell culture system) previously developed could only accommodate several thousand hepatocytes [6], making it difficult to do downstream assays such as RNA extraction and immunofluorescence staining. Since the flow distributor may expand the area at the membrane filter part, the cell seeding membrane below should also be designed at the same area so as to accommodate enough cells for each chamber.**Controllable Compaction**: In the report from Wang et al. [9], the compaction generated by the pressure applied on the ceiling of the elastic microchannel may positively modulate the cells inside. The compaction described in their paper, however, was performed by manually controlled tweezers. Here, the perfusion flow regulated by the external pump would also generate compaction force, but in a controllable way. When the flow resistance increases through integrating the micro-pore membrane at the outlet of the flow path, the pressure will build up until it is high enough to drive the perfusate through the nano pores. This pressure will then act as the compaction onto the cells cultured inside the chamber, in an even and controllable way. In our previous study, we have demonstrated that this compaction enhanced cell polarity and functions [11].**Efficient mass delivery**: The vertical-flow design allows all the media to pass through the cell culture membrane, enabling better media delivery to the cells. In order to supply enough O_2_ in the perfused media, a gas permeable tubing was used as the O_2_ generator before the perfusate reaches from syringe pump to the bioreactor. Specifically, in a drug testing application, it is important to avoid adsorption by the ECM (extra-cell matrix) overlay when adding the drug onto the cells in culture. Since the 2D cell culture setup in VfB may result in a similar morphology seen in 3D culture and in vivo, no overlay of ECM on top of the cells is needed, due to the compaction culture condition [11].

Additional features of a VfB would include:
**Biocompatibility**: All wet parts which comes into contact with culture media or cells were made from biocompatible materials. The main body is constructed with thermoplastic polymethyl methacrylate (PMMA), which has been reported to be compatible with various cell types including hepatocytes [22,23]. The VfB adopted commercially available Transwell® Polyester (PET) membranes as the cell seeding substrates, which are proven to be cell friendly and used widely for cell culture.**Scalable Throughput**: For drug testing purposes, a multi-well design is preferred for the purpose of high throughput. The VfB shares the same design culture chamber unit as a module, which enables flexible throughput from a single well, to double, 24 wells or higher. For instance, in order to scale up to a 24-well culture plate footprint, we could fabricate 6 groups of 4-well units to combine as an array [11]. The VfB is therefore flexible in throughput, and able to cater to various scale requirements in product design.

### 3.2. Device Fabrication and Thermo-Bonding Optimization

The main body of VfB was fabricated from PMMA plates machined according to the designed features (Figure 2F). As shown in Figure 1B, there were two bonding processes required for each VfB: two layers of PMMA sandwiched with PET membranes as the cap, and another two layers thermo-bonded without membranes as the bottom. As PMMA was proven to be a biocompatible thermoplastic, it is advantageous to use thermo-bonding as it does not involve adhesives or tapes, lowering the chances of complications from introducing foreign materials, such as micro-feature blockages, surface property, and biocompatibility changes, etc. [24]. In the later stage of scaled-up production, the simplicity and reliability of thermo-bonding would also enable the low cost and high yield manufacturing of the microfluidic device [23].

In this paper, we have tested two methods of thermo-bonding to assemble the VfB. The first one involved applying high pressure (as high as 200 psi) on PMMA via metal clamping plates and pressurized hotplates (Figure 4A). The bonding surfaces are required to be plasma treated after wiping down with isopropyl alcohol (IPA) and N2-blow drying. As the glass transition point (Tg) for PMMA is 105 °C [25], the temperature of the heating plate was adjusted from 93 to 96 °C, and the bonding time was 20–40 min based on the observation and adjustment for the different batches of PMMA materials. The processing time was short, but there were limitations on the bonding area and evenness, and the use of high pressure in bonding the VfB meant that the accurate temperature control was critical to avoid over- or under-bonding. Optimizations were needed for each batch, and the bonding effect was highly dependent on the operator’s skill. When under-bonding occurs, the parts may still be salvaged by increasing the initial 15-min bonding time by 10–15 min at 1–2 °C higher than the initial temperature set at 95 °C. When over-bonding occurs, the collapse of micro-features and the blocking of the shallow channels and chambers may largely change the flow resistance of the filters, causing the whole part to become unusable. The whole process involves much estimation and trial-and-error, and was heavily dependent on manual operation and observation through the naked eye. This method is therefore only suitable for rapid prototyping.

An improved method used was to gently apply pressure (via paper clips onto non-bonding clamping plates like thick glass) onto the PMMA layers and put the bonding parts into a high temperature oven to allow bonding at 120 °C (Figure 4C). A ready-to-use temperature protocol is shown in Figure 4D. As the bonding temperature is above the Tg of PMMA at 105 °C, the flowing of the material at the interface makes the bonding strength strong enough to hold the sandwiched PET membrane, even though PMMA and PET do not bond with each other. Besides, the space of a high-temperature oven allows for a much bigger bonding area and higher throughput than pressurized hotplates. The finished products were perfectly transparent with no visible delamination or defects (Figure 4E). Notably, the oven bonding required the bonding surface to be vacuum dried at 80 °C overnight (Figure 4B). If not done, the evaporation of adsorbed moisture may result in an uneven finish, with bubbles trapped on the bonded surface (Figure 4F). This mode of production is ideal for small scale production when the design and fabrication protocols are confirmed.

### 3.3. Prepare VfB for Cell Seeding

The VfB was connected with PEET tubing and four-way valves (Upchurch Scientific, IDEX Corporation, Lake Forest, IL, USA) through a machined PMMA jig. Silicone O-rings were used to couple and seal the hard surfaces between the VfB and jig, so that the perfusate can be delivered from a syringe to VfB. Notably, there is a silicone air-permeable-water-proof tubing (Silastic^®^ Laboratory tubing, 7-5224 Cat#508-009, Cole-Parmer, Vernon Hills, IL, USA) in the perfusion path, which serves as the O_2_/CO_2_ exchanger when the perfusion culture was done inside the CO_2_ incubator. The abovementioned parts were then prepared for the cell seeding procedures as shown in Figure 3.

In order to sterilize VfB for mammalian cell culture, the tubing, valves, and other PEET parts were autoclaved at 105 °C for 30 min and kept wet until used. Other bioreactor parts made from PMMA and silicone should not be autoclaved, but rather soaked in 70% ethanol overnight and then air dried on autoclaved tissue inside a UV-sterilized biosafety cabinet (BSC). On the day of cell seeding, the sterilized flow divider and culture layers were assembled inside the BSC, in the jig connected with perfusion tubing and with silicone O-rings used as water-proof sealing (Figure 3C).

The sterilized VfB with perfusion path needs to be primed before seeding. First, the whole flow path needs to be rinsed with a continuous perfusion of phosphate buffer saline (PBS) in the volume of at least 3 mL/well. This is to remove leftover water or ethanol from the sterilization steps, as these are hazardous to hepatocytes [26]. The PBS was then replaced by 0.2% bovine serum albumin (BSA)-PBS solution for 1 mL/well and left in the flow path with valves closed for >1 h. Immediate before cell seeding, the BSA-PBS solution was replaced with fresh William E medium for rat hepatocytes (Rat-Hep-WE). The VfB was then ready to accept the cell seeding membranes.

The PET Transwell^®^ membrane with the track-etched 0.4 µm pore membrane (Corning 3450) was chosen as the cell seeding membranes for its transparency. When taken off inside the BSC with sterilized blades, the membrane was kept larger than 16 mm so as to leave enough space for fixing onto the VfB. Coating of the membrane with extracellular matrix (ECM) facilitates the cell attachment. Here, we used the acidic collagen coating method with the briefed steps as below:Prepare 1:10 dilution of collagen gel stock in 0.1 N HCl at room temperature to achieve 0.3 mg/mL acidic collagen.Soak PET membrane in acidic collagen with a volume of 1 mL/well in 12-well plate. Make sure the whole cell seeding areas are immersed.Keep the 12-well plate in 37 °C incubator for 2 to 3 h.Wash three times with 5 min interval in 2 mL/well PBS in order to remove acidic solution thoroughly.Replace PBS with 1 mL/well Rat-Hep-WE and keep the membrane wet until prior to cell seeding.

The coating needs to be freshly done before cell seeding, and the coated membranes fixed onto the primed cell culture wells with sterile ring clips inside the BSC so as to get the VfB ready for cell seeding (Figure 3B). After seeding was completed, the cap of VfB was closed immediately; then the whole VfB was encased in the jig and connected with perfusion tubings and syringes (Figure 3C,D), and moved to a CO_2_ incubator for subsequent culture.

### 3.4. Optimization of Cell Seeding Density in the VfB

During in vitro culture, the hepatic functions and cell number of hepatocytes tend to drop when the culture time was extended, especially in perfusion cultures, where the risk of cell loss is higher than that in static cultures due to the introduction of the culture media flow [27]. Bioreactor designers tend to increase the density of cell seeding to compensate for hepatic function and cell loss. However, we believe that the optimum density should be optimized against hepatic functions, as unnecessary high seeding density does not only increase the cost of tests but may also incur negative impact on cell performance. Since urea production is an important hepatic function, which is easily quantified by sampling the perfusate on a daily basis without affecting the culture process, we used it to optimize the cell seeding density in each well of the VfB (Figure 5).

The hepatocytes cultured in the VfB with different seeding densities were compared, together with the static control which was seeded on the same type of PET membrane in a monolayer and cultured in a 24-well culture plate. As shown in Figure 5, hepatocytes formed colonies in all the cultural conditions we tested, with different morphology observed at the end of the 4-day culture: for seeding density at 0.2 and 0.4 million cells/well, a monolayer of colonies appeared in both static and compaction culture; while 0.8 million cells/well in compaction culture showed severe clumping of cells on the membrane with mixed cell morphologies, and those in static culture showed peeling off of cells from the membranes during the daily media change (picture not shown). When the hepatic function of urea production was quantified daily, all compaction cultured hepatocytes produced higher urea than those in static culture, while the 0.2 million cells/well showed the highest urea production rate. Therefore, the seeding density of 0.2 million cells/well was regarded as the optimized condition and was used for the following studies.

### 3.5. Hepatic Functions Were Enhanced to Facilitate Drug Testing Applications

For drug testing applications, it is preferable to maintain cell functions in the sustained culture so that the sub-chronic or chronic effect of drugs could be tested. In our previous work, the compaction culture was already shown to preserve the hepatocytes’ morphology after 14 days [11]. In this paper, we have observed that the urea and CYP functions of hepatocytes cultured in the VfB could be maintained and enhanced for up to 14 days, using the static culture as a control (Figure 6).

For urea production, the hepatocytes in VfB maintained at 8 to 20 folds higher than the static culture throughout the whole 2-week culture period (Figure 6A). For the CYP1A2, 2B2 and 3A2 results of the cells harvested at the end of the 2-week culture, the perfusion cultures all exhibited higher production levels than the static control as well (Figure 6B–D). All the quantification results here show that the VfB enhanced the hepatic functions of the hepatocytes for a long-term culture, which may be a promising drug testing platform.

## 4. Conclusions

Bioreactors for in vitro cell culture are useful tools for both scientific research (e.g., disease modeling, tissue engineering) and clinical applications (e.g., artificial organs, cell therapy, drug testing) [1,2,3,28,29,30]. An ideal bioreactor should abide by certain guidelines to provide a suitable microenvironment for cells to maintain their viability and functions throughout the culture period. This includes ensuring efficient mass delivery to support a sufficient number of cells growing inside the bioreactor, providing in-time media replenishment, supplied evenly to all the cells, mimicking the environment in the body for 3D and compaction culture, etc. [11]. In conceptualizing and designing bioreactors as a product for future industrialization, the scalability, manufacturability, biocompatibility, and ease of sample retrieval are all important considerations. Since our previously published vertical-flow compaction bioreactor array (VCBA) has been proven to meet the abovementioned guidelines [11], we seek to elaborate on the methodology and principles of design, setup, optimization, and test of the vertical-flow bioreactor (VfB) in this paper, using the double-well VfB as a case study.

In order to meet the requirements of an ideal bioreactor, the design principles were carefully established, such as the vertical-flow perfusion to ensure mass transfer, the flow enlarger and distributor to expand the effective cell culture area, the modular culture unit design for easy scale-up, etc. The components were carefully chosen, such as the micro-pore membrane as the shear stress membrane to protect the fragile hepatocytes cultured underneath, the biocompatible thermoplastic PMMA as the main construction material, the transparent cell culture membrane with reversible assembly to allow convenient retrieval for downstream assays such as immunostaining, LC/MS, cell lysis, and PCR, etc. The fabrication process was discussed, with an elaboration on the thermo-bonding process optimization to form a strong sandwich structure with PET membrane filter to bear the elevated flow pressure generated inside the culture chamber, which was then used as the compaction force for hepatocyte culture. The bioreactor prototype proved to enhance the hepatic functions of the cultured hepatocytes over sustained culture and showed promise in commercialization with scalable throughput as a manufacturable product.

## Figures and Tables

**Figure 1 biosensors-11-00160-f001:**
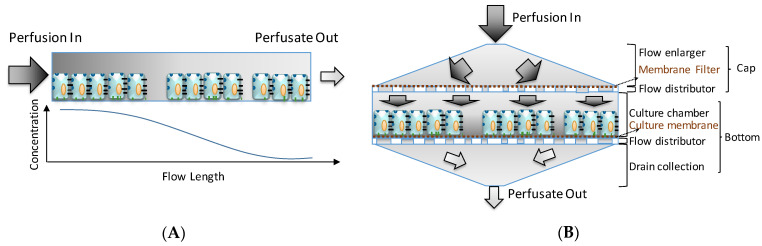
Horizontal- versus vertical-flow: (**A**) Horizontal flow brings an intrinsic gradient concentration when there is an absorption by the cells along the perfusion path. (**B**) Vertical-flow bioreactor distributes the perfusion evenly across the culture area, and makes the perfusate reach the cells on the same plane simultaneously with the original media concentration. (Illustration only, not draw on scale).

**Figure 2 biosensors-11-00160-f002:**
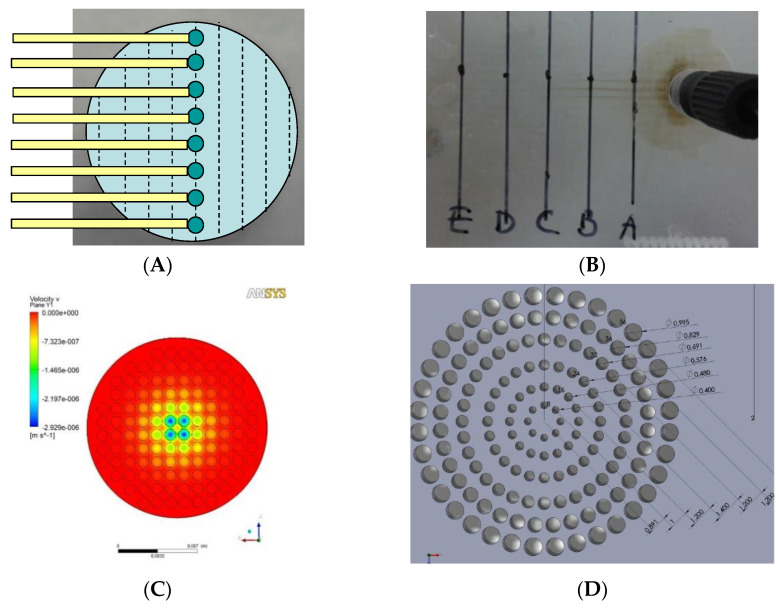
Flow distributor optimization: (**A**) Schematic design of testing chip which has sample collection channels starting along the diameter and was placed below the distributor to collect the perfusate. (**B**) In the case when hole size and positions are evenly distributed on the flow distributor, perfusion velocity was shown to be higher through the central sample holes than the periphery ones. (**C**) Simulation of flow velocity near the exit of the even-hole distributor at the overall flow rate of 0.1 mL h^−1^. (**D**) Dimension of finalized flow distributor design. (**E**) Simulation of flow velocity near the exit of the finalized flow distributor. (**F**) Layer structure of the VfB (double well design as the prototype described here).

**Figure 3 biosensors-11-00160-f003:**
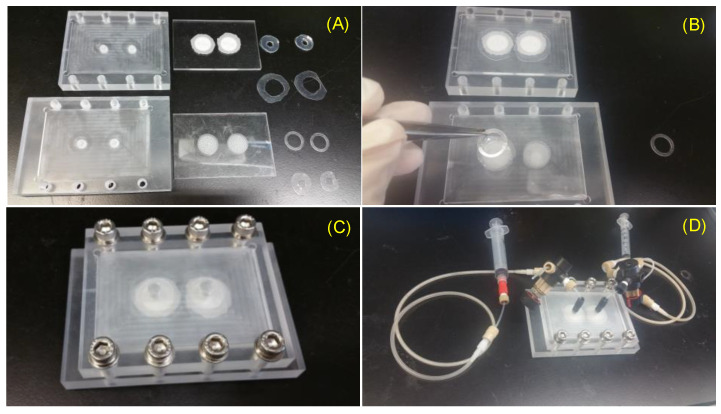
Assembly of VfB: (**A**) The VfB components: left column: jig to encase VfB; middle column: VfB cap and bottom; right column: O-rings to seal jig-cap, O-rings to seal cap-bottom, ring clips, O-ring to seal bottom-jig. (**B**) Fix a cell seeding membrane with a ring clip. (**C**) VfB encased in jig with screw security. (**D**) Connect perfusion tubing to VfB. Scale bar = 1 cm.

**Figure 4 biosensors-11-00160-f004:**
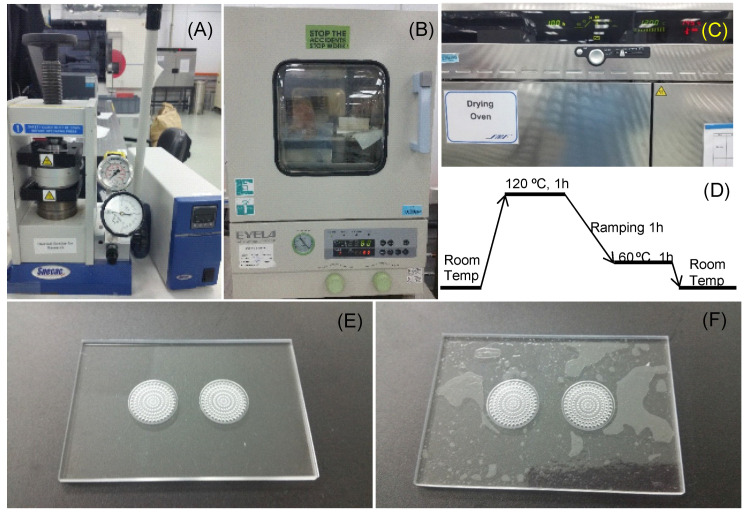
Thermo-bonding of VfB: (**A**) Thermo-bonding machine by hydraulic pressure. (**B**) Vacuum oven for surface dehydration. (**C**) High temperature oven for thermo-bonding. (**D**) Optimized bonding protocol in high temperature oven. (**E**,**F**) The finishing surface of thermo-bonded VfB with (**E**) and without (**F**) overnight vacuum oven dehydration. Scale bars = 1 cm.

**Figure 5 biosensors-11-00160-f005:**
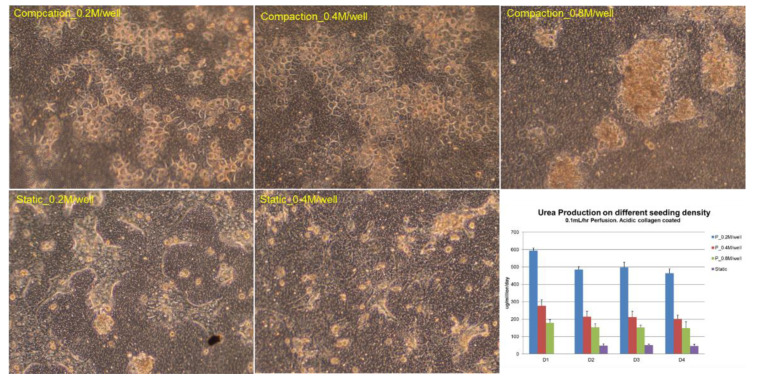
Cell seeding density optimization: Urea production was tested for 4 days for three different seeding densities: 0.2, 0.4, and 0.8 million cells/well, in both compaction (in VfB) and in static culture (in 24-well culture plate). All quantification results normalized to seeded cell numbers. Error bars show the standard error of the mean (SEM, *n* = 2–4). Scale bars = 50 μm.

**Figure 6 biosensors-11-00160-f006:**
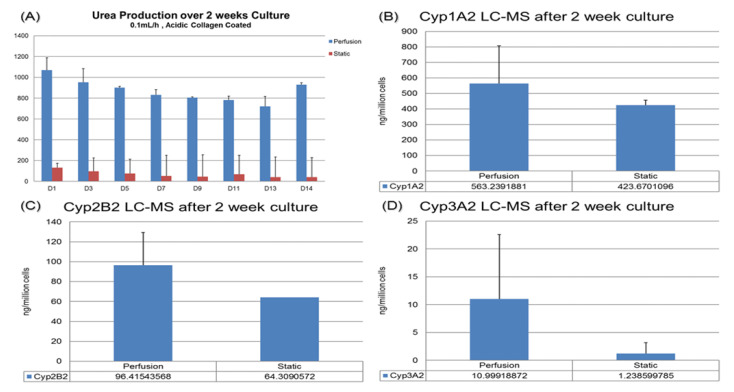
Hepatic function assays targeting drug testing application: (**A**) Urea production over 2 weeks. (**B**–**D**) Basic CYP level tested via LC-MC. Cells were harvested after 14-day culture: (**B**) CYP1A2. (**C**) CYP2B2. (**D**) CYP3A2.

**Table 1 biosensors-11-00160-t001:** Considerations in VfB to meet with the ideal features of cell culture device and bioreactor.

Ideal Features of Cell Culture Device for Drug Testing	Corresponding Considerations in VfB Design
Biocompatible, nontoxic material [14]	PMMA as main body, PET as cell culture substrate, with collagen coating
Optically transparent [15]	Transparent cell culture membrane for staining and imaging
Large surface area for enough cells to attach [14]	Cell culture chamber same as 24-well plate (Φ15 mm)
Efficient mass delivery [14]	Vertical-flow to ensure all media flow onto cells No ECM on top of cells to avoid additional adsorption of drug or media
Easy to assemble, insert and retrieve culture parts [15]	Open-cap design, allowing easy retrieval of cells and culture substrates
Flexible configuration: allow co-culture [15]	Allows for co-culture in mixture or different side of cell culture membrane
Scalable or High throughput [15]	Modular structure of culture wells and perfusion channels for easy scale-up
**Ideal Features of Perfusion bioreactor for drug testing**	
Closed culture environment [16]	Closed culture well during perfusion by embracing VfB inside the jig
Leak proof [15]	Seal rigid surfaces with silicone rings and fixed in jig via screws
Allows use of different flow types/rates [15]	Flow type/rates fully adjustable by syringe pump
No air bubble formation [15]	Simple structure without micro-channel features
Even distribution of flow across the large culture area *	Flow enlarger + Flow distributor to dissipate flow of media
Low buffer volume for in-time media replenishment *	Shallow culture chambers with vertical flow to efficiently replenish media
Minimize the shear stress *	Micro-pore membrane filter to filter shear stress Vertical flow enlarged from small to big cross section area (Φ1 to 15 mm)
**Biometic Features**	
Maintain 3D cell shape as in vivo [14]	Cells maintain 3D cuboidal shape with minimum stress fibers [11]
Controllable Compaction *	Cell compaction similar to intra-abdominal pressure (IAP) [11]

* the unique features achieved by VfB.

## Data Availability

The data presented in this study are openly available in https://scholarbank.nus.edu.sg/handle/10635/124178 (accessed on 1 May 2021).

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
