# Peer review of "Design and Fabrication of the Vertical-Flow Bioreactor for Compaction Hepatocyte Culture in Drug Testing Application"

_biosensors, 2021, doi:10.3390/bios11050160_

Round 1

Reviewer 1 Report

This manuscript optimized the design of vertical-flow bioreactor and achieved to maintain the 3D morphology of hepatocyte in a 2D setup. This methodology has the potential to provide a high-throughput manner for hepatotoxicity screening of drug candidates. Overall, this manuscript would justify a publication in Biosensors after the following concerns are addressed.

1. The labeling of subsections of “section 2. Materials and Methods” maybe changed to 2.1, 2.2, and 2.3 to maintain the consistence.

2. The Figure below the legend of Figure 1 is not described (Line 415)

3. The resolution of Figure 5 needs to be increased. The quantification data is not distinguishable.

4. Significant difference between different groups is not given.

5. There are some typos, such as Line 127 “[11] (Error! Reference source ot found.)”

Reviewer 2 Report

This manuscript details the method that the authors used for compaction culture of hepatocytes in 2016 (ref[11]). While this protocol-type manuscript is useful, the reviewer is not convinced that it matches the scopes and aims of Biosensor. Other comments,

  1. What is the thickness of the bottom PMMA layer as it seems quite thick. This would lead to difficulty in observing cells as the objectives with higher magnification powers cannot reach the cells due to their limited working distance.
  2. Introduction needs transition from the 2nd paragraph to the last paragraph.
  3. The quality of all figures is too poor. Details are not visible.
  4. A number of reference errors (e.g., “Error! Reference source not 173 found.”) should be corrected.

Reviewer 3 Report

The following comments may help the authors to improve the manuscript:

  1. PMMA is chosen in this study. The authors mentioned the possibility to scale up the fabrication. Perhaps, including the techniques to scale up, such as injection moulding, etc. See reference: https://doi.org/10.1016/j.trac.2020.116004. Also, other material such as COC, COP, PS should be discussed as an alternative.
  2. Line 127: ‘’microenvironment for cells [11] (Error! Reference source 127 not found.)’’
  3. Please include scale bars in figure 3, 4, and 5
  4. The simulation code or file should be included in the supplementary material for the readers to follow.

Round 2

Reviewer 2 Report

Previous concerns have been addressed. 

Reviewer 3 Report

There are some typos in reference 12, in the authors' name, in the revised manuscript.

The authors have addressed other comments.